# Instructing children to construct ideas into products alters children's creative idea selection in a randomized field experiment

**Kim van Broekhoven**[1]*, **Barbara Belfi**[2], **Lex Borghans**[3]

**1** Radboud Teachers Academy, Radboud University, Nijmegen, The Netherlands, **2** Research Centre for Education and the Labour Market, Maastricht University, Maastricht, The Netherlands, **3** School of Business and Economics, Maastricht University, Maastricht, The Netherlands

* kim.vanbroekhoven@ru.nl

**Data Availability Statement:** The data that support the findings of this study are publicly available in the Center for Open Science (OSF) at https://osf.io/thnyu/. DOI 10.17605/OSF.IO/THNYU.

## Abstract

Many popular pedagogical approaches instruct children to construct their ideas into tangible and physical products. With the prospect of implementation, do children decide to go for the most creative ideas or do they shift towards ideas that are perhaps less creative but easier to construct? We conducted a field experiment to test whether expected construction affects children's creative idea selection. In this experiment, 403 children were asked to select the most original ideas to make a toy elephant more fun to play with. We randomly assigned them to a treatment condition—in which they were informed they had to construct one of the original ideas that they selected—and a control group—in which children were informed that, after idea selection, they had to perform another task. Children who were instructed to construct the selected idea into a tangible product turned a blind eye to original ideas and preferred the more feasible ideas. Thus, pedagogical approaches that aim to stimulate creativity by instructing children to construct original ideas into tangible and physical products may unintentionally change children's choices for creative ideas. This finding highlights the importance for educators of guiding children's decision-making process in creative problem solving, and to be aware of children's bias against original ideas when designing creative assignments for them.

## Introduction

To develop children's creativity, constructivist pedagogies have risen dramatically across primary schools (e.g., Montessori education and project-or research-based learning [1]). Constructivism emphasizes the importance of the learner being actively involved in the learning process, and it has been established by educational science that children learn better when they develop external representations or products of their constructed knowledge [2, 3]. As such, an important characteristic of constructivist pedagogies is that children conclude their projects by reflecting their understanding, knowledge, and ideas in the construction of a final and concrete product, such as a prototype [4–6]. Further, the resulting products are popular means to assess creativity as they may be seen as the universal language of all children, irrespective of

**Funding:** The authors received no specific funding for this work.

**Competing interests:** The authors have declared that no competing interests exist.

their literacy skills, age, nationality, or intelligence [7–9]. The prospect of having to build a product based on an original idea might, however, make children hesitant to select such ideas, because it may be more of a challenge to actually build them; this may be detrimental for children's creativity development. To investigate this, we tested whether instructing children to construct original ideas into tangible products alters children's creative idea selection.

In this study, creativity was conceptualized as a set of specific characteristics of ideas that children selected, and these characteristics include originality and usefulness [10]. Originality refers to the characteristic of an idea that is new and unusual, and usefulness refers to the characteristic of an idea that is potentially feasible or valuable [11]. Originality is seen as the most important aspect of creativity because something must be original in some way to be considered creative [12]. While psychological and educational studies on *creativity* focus mainly on the generation of original and useful ideas [12], management and business studies on *innovation* focus mainly on the implementation of such ideas [13]. The present study takes a multidisciplinary perspective by linking these two approaches, acknowledging that creativity starts with generation of ideas that are then typically implemented. Yet, to move from creativity to innovation, the most creative ideas must be selected, and the prospect of implementation itself may affect idea selection. The terms implementation, construction, and building are used interchangeably in this article. This leads to the question of how the expected implementation of ideas affects children's idea selection.

Several findings in the social psychology literature suggest that extrinsic constraints, such as expected evaluation of ideas, hinder people's creativity [e.g., 14–17]. This may be so for several reasons. First, it has been argued that exerting external pressure and constraints may reduce the fun of performing a task, which in turn may reduce intrinsic motivation and, subsequently, creativity [15]. This line of research has shown that people are more likely to produce creative work when they are intrinsically motivated, because they are free of extraneous concerns about contextual conditions and are able to concentrate their attention to the task itself [16]. Arguably, the expected implementation of ideas is a form of extrinsic constraint as well, because children observe each other's attempts to build ideas into concrete products. Consequently, they may fear building unorthodox ideas that may fail or be ridiculed by their peers. Hence, expected implementation may exert external pressure on children's attempt to implement an idea in practice, and subsequently on their idea selection. Further, it has been found that people have a deep-seated desire to maintain a sense of certainty and to preserve the familiar [18]. In idea selection, people often favor the more common ideas due to risk perception with creative ideas, as creative ideas are by definition uncertain, because they are often new and untested [19]. According to the novelty-usefulness trade-off, creative ideas are often original, and the more original an idea is, the higher the risk perception whether an idea will work in practice [20], creating doubt as to whether the idea can be realized [12]. As such, children may fear failure in building physical products out of original ideas. For these reasons, simply instructing children to select original ideas—without having to build their chosen idea into a tangible product—may already be a challenging task. We further theorize that this may become even more problematic if teachers additionally instruct children to build original ideas into tangible products, because this invites the question of whether their idea can actually be built in practice.

Based on previous literature, we hypothesize that: (i) children who are instructed to implement ideas are more likely to reject highly original ideas, and (ii) children select more feasible ideas.

A prior laboratory experiment shows that another extrinsic constraint, expected evaluation, caused undergraduates to make their ideas more feasible [17]. While laboratory experiments elicit the pressure of extrinsic constraints, they generally have a low ecological validity, because

the experiment is done in an artificial environment. As such, field settings allow a more realistic setting where extrinsic constraints are elicited in the actual classroom. Therefore, we ran a field experiment in which children in their natural school environment were not only asked to select the most original ideas of a set of ideas, but were also asked to actually build their selected ideas after selection. Research in developmental psychology has shown that children aged 10 to 13 show an increasing degree of conformist thinking that continues through high school [21–23]. Due to this increasing conformist way of thinking, the consequences of extrinsic constraints such as actually constructing an idea into a concrete product could become even more detrimental for children's creativity in a natural school classroom environment (where children observe each other's attempts to build ideas into concrete products).

## Methods and materials

### Participants

Before data collection, a priori power analysis revealed that a minimum of 388 participants would be required to obtain a statistical power of 0.99 with a (independent) t-test [G*Power 3.1; 24]. We recruited slightly more participants to compensate for drop-outs due to potential technical issues (e.g., problems with internet connection in the school). Data were collected from 403 children from 13 primary schools in the Netherlands between February and June of 2019. The children (49.9% girls) attended grade 6 (last grade of primary school) and were aged between 10 and 13 years ($M$ = 11.6, $SD$ = 0.48).

This study was reviewed and approved by the Ethical Review Committee Inner City Faculties from Maastricht University (ERIC_090_14_06_2018). Families of children in participating schools received a letter describing the project, and their consent was obtained through passive consent wherein they had the opportunity to ask for their child not be included in the project. The research team received withdrawal from participation requests for two children.

### Procedure

For our field experiment, children filled out an online assignment in their natural school environment as part of their daily school program (see S1 Appendix in S1 File for online assignment). Both the teacher and a researcher—who had been introduced as a substitute teacher—were present. The assignment consisted of three tasks and took in total 20 minutes to complete (see Fig 1 for experimental design). Both the researcher and the teacher walked around to answer questions about possible ambiguities.

In the online assignment, children were first asked to evaluate 20 ideas from a pre-defined list in terms of feasibility and originality to improve a stuffed toy elephant, such as enlarging the toy elephant or creating a toy elephant that is able to spit fire. Next, all children were asked to generate as many ideas as possible for toys for monkeys in the zoo. After this task, children were randomly assigned to either the experimental condition or the control condition. Children in the experimental condition (N = 201) were told, through written instruction on the computer screen, to expect future implementation of their selected ideas:

> "A toy factory needs your help! The toy factory makes toy animals, such as elephants, dogs, rabbits and so on. They would like to receive original ideas to change a toy elephant. They will first test these ideas on a toy elephant made of paper. **You will build these ideas.**"

In contrast, children in the control condition (N = 202) were told:

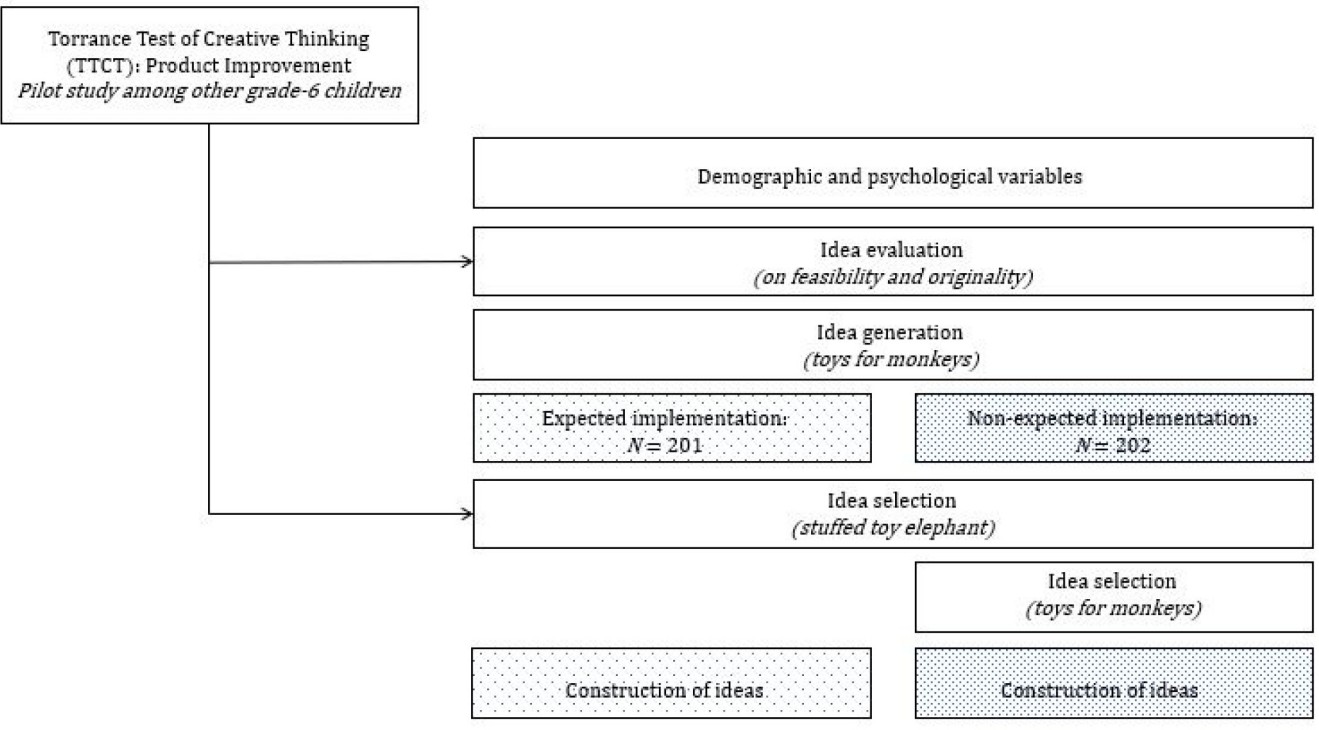

**Fig 1. The experimental design.**

"A toy factory needs your help! The toy factory makes toy animals, such as elephants, dogs, rabbits and so on. They would like to receive original ideas to change a toy elephant. They will first test these ideas on a toy elephant made of paper. **You will <u>NOT</u> build these ideas, because you will be building ideas for monkey toys.**"

After reading these instructions, the children were asked to select five original ideas to improve a toy elephant. From these five ideas, they had to select the two most original ideas. Thus, the manipulation was that some children were told that they later had to implement (i.e., build) these ideas, while other children were told that would not have to implement these ideas. The control condition received an additional task to select ideas to build a toy for monkeys in the zoo (see S1 Appendix in S1 File for complete materials for both conditions).

Teachers were provided with building materials for the children. This pack of materials included colored pencils, paper, scissors, glue, foam balls, magnets, iron wire, string, wool, paper clips, water, bouncy balls, bandages, plastic straws, tubes, and bags. At the end of this online assignment, children had to build their selected ideas (see Fig 2 for two examples of final products for the stuffed toy elephant). After this building exercise, the experiment ended and the children went back to their normal school program.

## Measures

**Idea pool.** As part of this study, 36 grade-6 pupils from a previous cohort had generated ideas to improve a stuffed toy elephant as part of the Torrance Test of Creative Thinking [TTCT; 25]. This product improvement task resulted in 438 ideas. This number was reduced to 369 by excluding ideas that involved non-play uses, such as make the elephant alive or use it as a pincushion. The ideas were then further reduced to a list of 62 ideas by excluding ideas

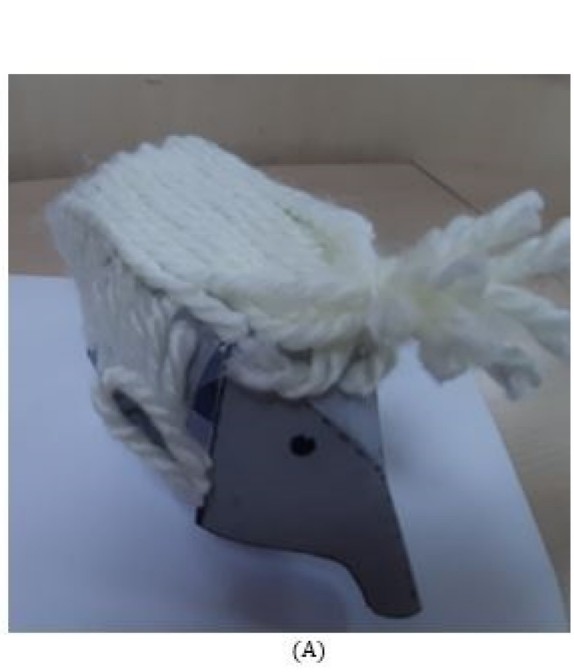
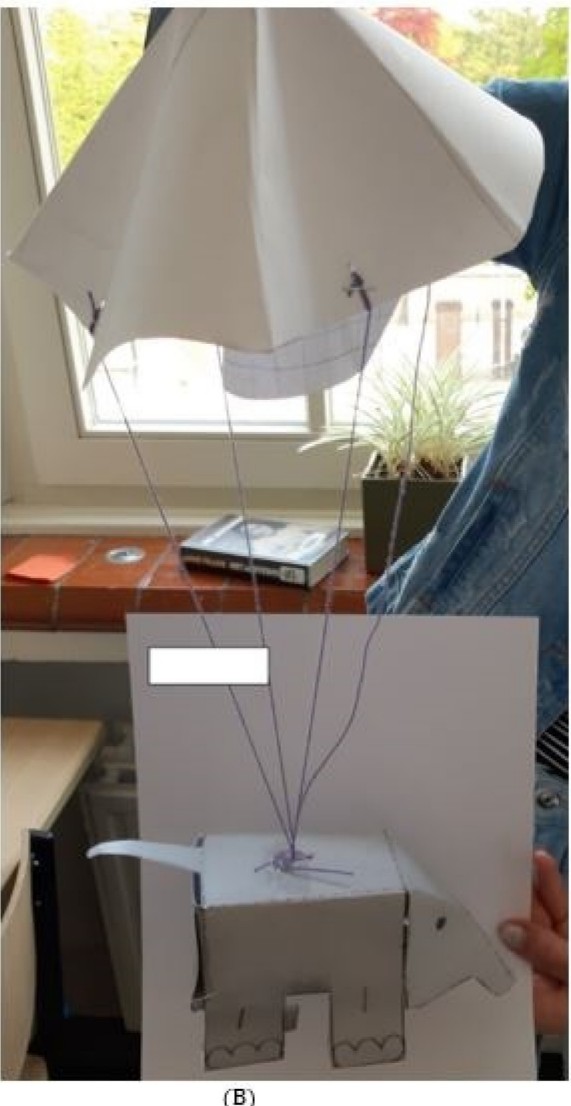

**Fig 2. Two examples of final products for the toy elephant.** Picture A presents the idea of 'make the elephant soft', and picture B presents the idea of 'make the elephant in such a way that it can fly'.

that were similar (i.e., the ideas "make it bigger" and "making a XL elephant" were collapsed into one idea, "enlarge the toy elephant"). Next, the remaining 62 ideas were rated by seven experts (i.e., four primary school teachers and three creativity researchers). The experts were instructed to rate each idea on feasibility and originality using a Likert scale ranging from 1 (not at all feasible/original) to 5 (very feasible/original). To reduce the list of 62 ideas even further, a random set of 20 ideas was selected to be presented to the children in the experiment (see S2 Appendix in S1 File). These 20 pre-defined ideas varied in creativity, in a 2 (originality: low, high) by 2 (feasibility: low, high), as a creative ideas has to be (a) original and (b) feasibility [10]. Interrater reliability for this list of 20 pre-defined ideas was high: The overall intraclass correlation coefficient (ICC, two-way random, consistency analysis) was 0.90, and the single interrater reliabilities were excellent (feasibility ICC = 0.94 and originality ICC = 0.86). By

averaging the scores of the seven experts, each single idea received a feasibility and originality score.

**Idea evaluation task.**    From this list of 20 pre-defined ideas for the stuffed toy elephant, children were asked to evaluate each idea on their feasibility (with a 5-point rating scale). Similarly as in Charles and Runco [26], children were asked to select one of five faces: (1) frown = *very difficult to build this idea*; (2) slight frown = *difficult to build this idea*; (3) no expression = *not difficult but also not easy to build this idea*; (4) slight smile = *easy to build this idea*; (5) smile = *very easy to build this idea*; that best showed what they thought of the idea. Next, the originality evaluation task was administered with a 10-point rating scale. The children were asked to estimate a hypothetical number of children (between 1 and 10) able to generate a given idea [26]. Hence, the originality of an idea was operationalized as the degree to which children thought that a number of children, from 10 children in total, would have generated a similar idea. This measure is often used in creativity research to determine the originality of ideas [26–31], and the pilot study showed that children were better able to assess the originality of ideas in this way.

**Idea selection task.**    After the evaluation task, children were asked to select the five most original ideas from the list of pre-defined ideas for the stuffed toy elephant. From these five ideas, they had to select the two most original ideas (original ideas were defined as ideas that children would rarely see). Based on children's evaluation ratings, children's idea selection performance is measured by children's own-rated feasibility and originality level of each idea. Next to children's own-rated feasibility and originality, we also have the average rating of the seven experts for feasibility and originality.

**Control variables.**    To test whether our findings are consistent across all children, we measured demographic and psychological variables that prior studies reported as relevant [32, 33]. Children's demographic variables (i.e., children's gender, age, ethnicity, socioeconomic status, prior level of achievement) originate from data from the Onderwijs Monitor Limburg. This is a large cooperative project between Maastricht University and schools, school boards, and local government. This data collection aims in particular to collect and analyze information about the educational development of students to foster educational improvement. The data contain school administrative data—concerning each child's gender, date of birth, ethnicity, parental educational level, and school site—and report grade for grade 6. Prior to the experiment, children's psychological variables were measured (i.e., risk preference and personality traits) in another online questionnaire (see S3 Appendix in S1 File). *Risk preference* was measured using the Risk Taking 10-item scale from the Jackson Personality Inventory [JPI; 34]. Children received adjusted statements suitable for children (presented in random order). Sample statements include: "I take risks" and "I like adventure." Children rated how well each statement describes themselves on a Likert scale, ranging from 1 (very inaccurate) to 5 (very accurate). Scale reliability (Cronbach's alpha) in this study was good ($\alpha = 0.84$). *Personality traits* were measured using the 50-item version of the International Personality Item Pool [IPIP; 35]. For each personality trait, children received 10 adjusted statements suitable for children (presented in random order). Sample statements include: "I am bursting with ideas" and "I am always prepared." Children rated how well each statement describes themselves on a Likert scale, ranging from 1 (very inaccurate) to 5 (very accurate). Scale reliability (Cronbach's alpha) in this study was good: openness to experience ($\alpha = 0.75$), conscientiousness ($\alpha = 0.80$), agreeableness ($\alpha = 0.74$), extraversion ($\alpha = 0.71$) and emotional stability ($\alpha = 0.82$).

## Data analysis

To investigate whether children select less original, but more feasible ideas, we needed both variables ranging from 1 to 5. For this, we transformed the originality rating for each idea

from the ten-point Likert scale to a five-point Likert scale. This was done by first dividing the originality evaluation ten-point Likert scale by two (range from 0.5 to 5). Next, we reversed these values so that higher values indicated higher originality (range from 1 to 5). Subsequently, we tested whether children's idea selection performance varied by condition in univariate ordinary least squares regressions, with children's average rating for the two selected ideas as outcome variables separately for feasibility and originality:

$$Y_{ija} = \beta_0 + \beta_1 \text{treatment}_{ij} + e_{ij}$$

where $Y_{ija}$ is the outcome of child $i$ in group $j$ ($j = 0$ for control, $j = 1$ for treated) and $a$ refers to the average feasibility or originality rating by child $i$ for the two selected ideas. As such, $Y_{ija}$ indicates children's average own rating of the two selected ideas, separately for feasibility and originality. The variable of interest, treatment$_{ij}$, is a binary variable that takes the value 1 for children who expected implementation of their selected ideas, and zero for children who did not expect implementation of their selected ideas. Lastly, $e_{ij}$ is a normally distributed residual with zero mean and constant variance $\sigma_e^2$ [e.g., 36–38]. As a robustness check, the average rating of the seven experts was also used to measure idea selection performance.

To test whether these findings were consistent across children's gender, age, ethnicity, socioeconomic status, prior level of achievement, and psychological variables (i.e., risk preference and personality traits), we performed multivariate ordinary least squares regressions with children's average rating for the two selected ideas as outcome variables separately for feasibility and originality, where we controlled for demographic and psychological variables:

$$Y_{ija} = \beta_0 + \beta_1 \text{treatment}_{ij} + \beta_2 D_{ij} + \beta_3 P_{ij} + e_{ij},$$

where $Y_{ija}$ is the outcome of child $i$ in group $j$ ($j = 0$ for control, $j = 1$ for treated) and $a$ refers to feasibility or originality. In addition to the univariate ordinary least squares regression, $D_{ij}$ refers to demographic variables (i.e., children's gender, age, ethnicity, socioeconomic status, prior level of achievement) and $P_{ij}$ refers to psychological variables (i.e., risk preference and personality traits).

Since there is a negative correlation between feasibility and originality, a reduction of the originality of ideas in case of expected implementation might be the direct result of increased feasibility. To check whether children really reduce the originality of the idea, more than would just be expected based on this correlation, we ran a conditional logit model that simultaneously estimates the effects of children's feasibility and originality ratings on their selection of ideas [39]. In this way, we partialize out the feasibility rating from the originality rating. In brief, each child chooses two ideas of a set of 18 ideas to improve a stuffed toy elephant. The probability that child $i$ chooses $k$ among $j$ alternatives is:

$$\Pr(i \text{ chooses } k) = \Pr(V_{ik} > V_{ij}) \; \forall j \neq k, j = 1, \ldots, J$$

In general, the utility of alternative $j$ for child $i$ is given by:

$$V_{ij} = x_{ij}\beta + v_{ij} \quad i = 1, \ldots, n; j = 1, \ldots, J$$

where $x_{ij}$ represents the variation of the feasibility and originality ratings across ideas. These ratings interact with the treatment (i.e., expected implementation or non-expected implementation). All variables must vary across ideas (or alternatives) to achieve identification in the conditional logit model. Therefore, as treatment is alternative-invariant, it can be included in the model only as an interaction with the characteristics of the alternative (i.e., feasibility and originality). Specifically, the interaction terms included are:

- *Treatment$_i$ * Feasibility$_{ij}$*

- *Treatment$_i$ * Originality$_{ij}$*

Results are reported as odds ratios. These should be interpreted as the proportional change in the odds of child *i* selecting idea *k* for a unit increase in the treatment variable, holding all other variables constant. This means that we can draw conclusions about the probability for children in the expected implementation condition and children in the non-expected implementation condition to select an idea given its feasibility and originality. It is important to be clear about what is meant by "change in the odds" in these models. This is based on the number of children making a particular choice (i.e., selection of two ideas) while accounting for the number of alternatives available within that choice set (i.e., 18 other ideas). Thus, when we say that the probability of selecting a feasible idea is higher among children in the expected implementation condition than children with no such expectation, this is after accounting for the total number of ideas in the total set.

## Results

The manipulation was successful: 93% of children in the expected implementation condition expected to construct ideas for a toy elephant in contrast to 10% in the non-expected implementation condition ($F(1, 211) = 466.49$, $p < 0.001$, Cohen's $d = 2.969$) (S1 Table in S1 File).

To investigate whether expected construction affects children's creative idea selection, we compared children's self-rated levels of feasibility and originality of their two selected ideas between the treatment and control group. We analyzed differences between the experimental and control group using ordinary least squares regression, and analyzed differences between the experimental and control group controlling for demographic and psychological variables as a robustness check (S2 Table in S1 File). In these analyses, degrees of freedom varied slightly across analyses because some children did not fill in the questionnaire containing the psychological variables. Furthermore, to understand children's trade-off between feasibility and originality when choosing a creative idea, we ran a conditional logical model that simultaneously estimated the effects of children's feasibility and originality ratings on their selection of ideas.

Instructional effects on creative behavior were found to be reliable and large in magnitude [40, 41]. Compared with those in the control group, children who expected implementation of ideas selected more feasible ideas. The Cohen's $d$ effect size is 1.028 ($B = 1.16$, $SD_{treatment} = 1.12$, $SE = 0.11$, 95% CI = 0.94 to 1.38, $\eta = 403$, $t(1, 401) = 10.31$, $p < 0.001$) (Fig 3). However, these children who did expect implementation selected significantly less original ideas. The Cohen's $d$ effect size is small to medium: 0.377 ($B = -0.43$, $SD_{treatment} = 1.23$, $SE = 0.12$, 95% CI = -0.66 to -0.21, $\eta = 403$, $t(1, 401) = -3.78$, $p < 0.001$) (Fig 3). These results are robust for the selection of five original ideas as well.

More specifically, the conditional logit models show that when taking a decision about the level of creativity of an idea, the expectation of idea implementation increased the probability of choosing a feasible idea by 103%, while the probability of choosing an original idea declined by 14% (Table 1).

Notably, the detrimental effect of instructing children to transform their ideas into tangible and physical products was consistent among children with different background characteristics (i.e., gender, age, ethnicity, socioeconomic status, prior level of achievement) and psychological characteristics (i.e., risk preference and personality traits with exception of conscientiousness). Hence, these findings are broadly applicable to children independent of demographic or psychological characteristics (S2 Table in S1 File). Further, results did not change when we used expert ratings as the outcome (S3 Table in S1 File).

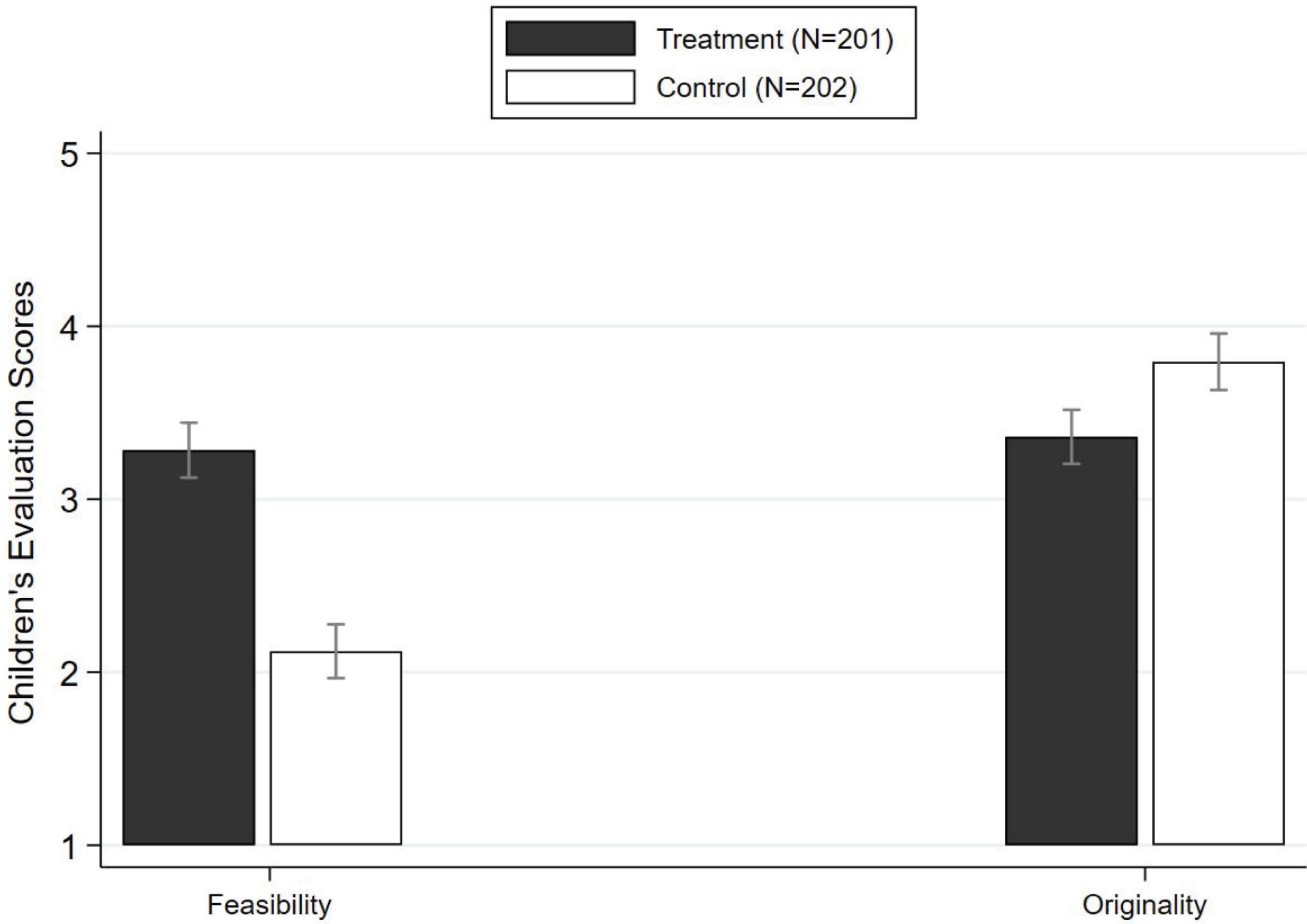

**Fig 3. Estimated effect of expected implementation on children's feasibility and originality ratings.** Error bars represent 1 SE. This figure summarizes the intervention's effect on children's feasibility and originality ratings. Error bars reflect 95% confidence intervals.

**Table 1. Idea selection conditional on alternative ideas (odds ratios and Z-statistics).**

|  | Model 1 |  | Model 2 |  | Model 3 |  |
|---|---|---|---|---|---|---|
| Feasibility | 0.604 | *** |  |  | 0.621 | *** |
|  | (-11.05) |  |  |  | (-10.33) |  |
| Treatment*Feasibility | 2.073 | *** |  |  | 2.026 | *** |
|  | (12.29) |  |  |  | (11.78) |  |
| Originality |  |  | 1.296 | *** | 1.202 | *** |
|  |  |  | (5.94) |  | (4.13) |  |
| Treatment*Originality |  |  | 0.771 | *** | 0.859 | * |
|  |  |  | (-4.41) |  | (-2.52) |  |

*Notes*: Z-statistics are reported in parentheses to indicate statistical significance. Effects are interpreted as the probability of favoring idea $k$ multiplied by a one-unit increase in that variable. Estimates greater than 1 are considered positive effects, while estimates smaller than 1 are considered negative effects.

*** Statistical significance at the 0.10% level (Z-statistics > 3.10)

** Statistical significance at the 1% level (Z-statistics > 2.58)

* Statistical significance at the 5% level (Z-statistics > 1.96).

## Discussion

Many constructivist pedagogies aim to develop children's creativity by instructing them to actually construct their ideas into tangible and physical products. Yet, little is known whether children decide to go for more or less creative ideas with the prospect of implementation. The aim of this study was to examine whether children select less creative ideas when they expect to implement these ideas on a later moment. More specifically, the current study investigated the effect of expected implementation of ideas on the selection of original and feasible ideas among primary school children.

We found that children inhibit themselves in selecting original ideas once there is an expectation of idea implementation. Hence, to move from creativity to innovation, this research shows that children may put more focus on the practicality of ideas. In a brainstorm between two people (i.e., dyad) about ways to improve a stuffed toy elephant, Glăveanu, Gillespie, & Karwowski [42] have compared the practicality of ideas recorded on paper with ideas not recorded on paper (but only verbally expressed). In line with our findings, they found that dyads were more likely to write down practical ideas than original ideas. As such, it seems that the practicality of ideas becomes more important in the selection and implementation of ideas than in the generation of ideas.

These findings are broadly applicable to children independent of demographic or psychological characteristics. We only found an exception for the personality trait conscientiousness. The significant interaction effect illustrates that more conscientiousness children tend to choose more feasible ideas, also when they are not in the treatment group (S2 Table in S1 File). Yet, their choice changes much less when they are assigned to the treatment group. Hence, untreated conscientiousness children show the behavior of others when they are treated.

Furthermore, we found a trade-off between novelty and usefulness where children who expected implementation selected less original, but more feasible ideas than children who did not have an expectation of later implementing the selected ideas. According to the novelty-usefulness trade-off, highly original ideas are more likely to be judged as less feasible because they involve, by definition, a step into the unknown [e.g., 43–48]. Several scholars argue that these two criteria of creativity are often seen as incompatible and represent a fundamental tension or paradox [e.g., 49–54]. In line with von Thienen, Ney and Meinel [55], we found a negative correlation between originality and feasibility of 0.51 among experts, and 0.36 among children. As such, the novelty-usefulness trade-off explains our finding that children select less original, but more feasible ideas once expecting implementation of those ideas. This implies that the expectation of idea implementation causes children to play it even safer with regard to practicality and to choose more feasible ideas rather than more original ones.

In addition, our finding can also partly be explained by the bias against originality [52]. The most original ideas are often those that are radically different from existing solutions or practices, which often cause people to have ambivalent feelings towards novel ideas, because people often prefer the status quo or familiar ideas. Blair and Mumford [56], for example, found that managers concerned with idea implementation prefer non-original ideas even when they are ideas that are both original as well as useful. We found that children without expected implementation selected more original ideas, they did not totally abandon feasible ideas.

### Practical implications

The results of this study have important implications for educational practice. First and foremost, our finding that children inhibit themselves in selecting original ideas once there is an expectation of idea implementation suggests that this instructional approach may lead to loss of potential original ideas. This means that by focusing on end-products only, educators may

run the risk of applauding the creativity of a small group of children who succeeded into constructing original ideas into tangible products, while at the same time not recognizing the creative potential of a group of equally creative children who tried, but failed in constructing original ideas into physical products. As such, educators should focus not only on the end-product, but particularly on the decision-making process children go through in selecting creative ideas [4]. In stimulating children's creativity, it may be worthwhile for educators to support children in their intuitive judgments of original ideas, in resisting peer pressures to conform, and guide children with highly original but seemingly unrealistic ideas to think of ways for the idea to be made feasible. There are several strategies that educators can use to render wild-sounding ideas into more useful or feasible ideas, and this may help children to pursue original ideas [e.g., 57–59]. More specifically, wild-sounding ideas can be rendered more effective by means of parallel prototyping [57]. For instance, educators could encourage children to imagine, try out constructing multiple ideas in parallel into tangible products. Another highly effective approach to render original ideas more effective is iterative prototype testing [57, 60]. In iterative prototyping, educators could encourage children to test and refine their ideas multiple times, moving from non-refined prototypes–such as rough sketches–to refined prototypes–such as refined paper models or CAD drawing over time. Next to these strategies, educators could also explicitly instruct children to try out at least one daring, wild "dark horse" solution. This 'Dark horse' strategy calls for the exploration of wild ideas that may never otherwise be explored, and ensures that highly original ideas do not get lost in the implementation phase [59]. For this, it is important for teachers to foster a psychologically safe classroom environment where all children feel safe in playing and experimenting with ideas and materials, taking sensible risks, and making mistakes. In such an environment, novel and unorthodox ideas are valued and failures are seen as a necessary and positive part of the learning process in support of creativity [61–63].

## Limitations and future directions

To the best of our knowledge, this is the first study demonstrating the effect of expected implementation on children's idea selection. To test this, a relatively large sample of 403 children were asked in a randomize design to select two innovative ideas with or without the expectation to having to implement these ideas in the classroom.

Despite its strengths, our study inevitably has limitations that future research may address. First, we investigated the effect of expected implementation on idea selection among children aged 10–13, because several studies show that children around this age begin a trend of increasingly conformist thinking that continues through high school [21, 22]. This manifests itself in the fact that children want to be as 'normal' as possible, and prefer to do everything the same as their peers. As a result, it is therefore often more important to them what children of the same age think of them and their behavior than their teachers or parents [64]. Accordingly, it can be expected that our findings may become even stronger among young adolescents. However, it remains unclear how younger children or adults would perform in similar experimental conditions. Thus, while the present study provides a starting point in research on the effect of expected implementation on creativity, the question remains as to how expected construction of ideas into tangible and physical products affects creativity in younger and older age groups.

We may also note that while this study investigated whether the expectation of idea implementation affects idea selection among primary school children, it did not investigate the underlying mechanisms why this happens (e.g., emotional aspects of idea selection). For instance, children's emotional reaction, such as fear of failure, to expected implementation

might explain why children have a bias against original ideas in their selection [65, 66]. Prior research has showed that people have a natural bias against creativity over feasibility because of uncertainty [15, 52]. As such, children may select different types of ideas to reduce uncertainty in the implementation phase [67]. Future research should aim to investigate these underlying mechanisms.

Finally, it is possible that the creativity of the children's final selected ideas was lowered, because children may be constrained by their own building capacity in their selection of original ideas [68]. Children may wonder which idea they can actually build instead of selecting original ideas irrespective of their building skills. Further, several researchers have shown that multiple iterations and parallel prototyping contributes to an increase in both the quantity and creativity of prototypes produced, as it allows children to try out constructing multiple ideas in parallel into tangible products [e.g., 57, 60]. Therefore, instead of giving children only one opportunity to translate their idea into a tangible product, children may gain more experience by letting them repeatedly work on a prototype in parallel sessions to further refine their idea and this may boost the creativity of their idea.

## Conclusion

In sum, the present study investigated the effect of expected implementation of ideas on children's selection of original and feasible ideas. The results showed that expected implementation exerted different effects on the two dimension of final product creativity. Children who expected implementation selected less original ideas, but more feasible ideas than did children in the non-expected implementation condition. Thus, pedagogical approaches that aim to stimulate creativity by instructing children to construct original ideas into tangible and physical products may unintentionally change children's choices for creative ideas. This finding highlights the importance for educators of guiding children's decision-making process in creative problem solving, and to be aware of children's bias against original ideas when designing creative assignments for them.

## Supporting information

**S1 File.**
(DOCX)

## Acknowledgments

We are grateful to all of the teachers and children who participated in our research. We thank Sylvie Beckers and Maurice Thelen for their assistance with data collection, and their expertise on how to integrate our experimental design in the grade 6-curriculum. Research reported in this publication was supported by the Educatieve Agenda Limburg and the Kindante board of primary schools.

## Author Contributions

**Conceptualization:** Kim van Broekhoven, Barbara Belfi, Lex Borghans.

**Data curation:** Kim van Broekhoven.

**Formal analysis:** Kim van Broekhoven.

**Investigation:** Kim van Broekhoven.

**Methodology:** Kim van Broekhoven, Lex Borghans.

**Project administration:** Kim van Broekhoven.

**Supervision:** Barbara Belfi, Lex Borghans.

**Visualization:** Kim van Broekhoven.

**Writing – original draft:** Kim van Broekhoven.

**Writing – review & editing:** Kim van Broekhoven, Barbara Belfi, Lex Borghans.

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
