## [Decision Letter · Decision Letter 0]

4 Apr 2022

PONE-D-22-04429Instructing children to construct ideas into products decreases creativity in a randomized field experimentPLOS ONE

Dear Dr. van Broekhoven,

Thank you for submitting your manuscript to PLOS ONE. After careful consideration, we feel that it has merit but does not fully meet PLOS ONE’s publication criteria as it currently stands. Therefore, we invite you to submit a revised version of the manuscript that addresses the points raised during the review process.

 I've now received comments by two experts. Both see merits in this work and appreciated the methodological and research design approach described in the paper. However, both reviewers highlighted specific concerns with the theoretical and interpretative approach of the work, which need to be addressed by the authors. After my own reading of the manuscript, I agree with the comments of the reviewers. In particular, I would suggest to integrate the introductory section with the abundant literature on implementation, as suggested by Reviewer 2. Moreover, a more precise and careful interpretation of the results is suggested to the authors. Both Reviewer included thoughtful and insightful comments and suggestions (Reviewer 1 as an attachment), which I strongly recommend to follow in the revision of the manuscript. 

We look forward to receiving your revised manuscript.

Kind regards,

Sergio Agnoli

Academic Editor

PLOS ONE

Journal Requirements:

Reviewers' comments:

Reviewer's Responses to Questions

**Comments to the Author**

1. Is the manuscript technically sound, and do the data support the conclusions?

Reviewer #1: Yes

Reviewer #2: Partly

2. Has the statistical analysis been performed appropriately and rigorously? 

Reviewer #1: Yes

Reviewer #2: Yes

3. Have the authors made all data underlying the findings in their manuscript fully available?

Reviewer #1: Yes

Reviewer #2: No

4. Is the manuscript presented in an intelligible fashion and written in standard English?

Reviewer #1: Yes

Reviewer #2: Yes

5. Review Comments to the Author

Reviewer #1: Thank you for the opportunity to review the text entitled "Instructing children to construct ideas into products decreases creativity in a randomized field experiment." In the field experiment presented in this article, 403 children aged 10 to 13 were asked to select the most original ideas to make a toy more fun to play with. A random half of children was told that after the idea selection, they are expected to craft the chosen ideas—to transform them into tangible products. The other half was informed that they are going to engage in another task after idea selection (so they did not expect to construct anything based on the selection made). The analyses shown that children who anticipated implementing the chosen ideas tended to pick more feasible yet less original ideas as compared to their peers who did not expect crafting the ideas. Using conditional logit models, the Authors shown that the probability of choosing a feasible idea increased by 103% and that of choosing an original idea declined by 14% when expecting crafting the ideas. These results were consistent irrespective of children’s demographic characteristics, their personality, and risk preference. Furthermore, the conclusions stayed the same with ideas’ originality self-rated by children (prior to idea selection) as well as when expert judges scored the ideas for originality.

Overall, I enjoyed reading this paper very much. It is clearly written, thought-provoking, and brings important message for both creativity and education research and practice. The performed analyses are elegant and well-suited for the research questions stated. The technical aspects of the presented study are of high quality making this work a nice fit for PLOS ONE. I have mostly minor suggestions for the Authors’ consideration that would make the paper even clearer and more understandable for the reader.

Please have a look at the attached review to see my specific comments.

Reviewer #2: This is a great study in terms of testing a large sample of children, with a clear-cut experimental manipulation and a good test battery to assess control variables. The topic is highly relevant, as creativity is an important learning objective at school, and the development of suitable education programmes is an enduring challenge.

To improve this paper even further prior to the final publication, a couple of suggestions follow.

First, there has been a lot of research on how implementation impacts ideation, and the outcomes of creative processes. Some key findings from this research should be integrated in the paper. Most notably, the kind of implementation that has been required from children in this experiment is a particular kind of implementation, which indeed would be expected to foster the opposite of creatively diverse and daring ideation. This form of implementation can be characterised as follows: (i) the children’s prototypes have a relatively high resolution, (ii) they require manual building skills, (iii) there is only one prototype construction round by the end of the process, (iv) there is no explicit use of multiple prototypes to try out diverse ideas including at least one “wild” approach. In all four regards, more creatively diverse and daring solutions would have been expected if the implementation process had been different.

(i) A well-replicated finding in creativity and innovation research is that of Edelman & Currano (2011) in their Media-Models Framework: Prototypes with a high resolution/refinement/granularity (e.g. CAD drawings, or refined paper models) foster convergent thinking, and little changes are made to already existing concepts. By contrast, rough prototypes (e.g., rough sketches or improvised, non-detailed 3D depictions) foster divergent thinking and invite deviance from existing concepts. The two sample elephants depicted in the present experiment show a notably high amount of refinement.

Ref:

Edelman, J., & Currano, R. (2011). Re-representation: Affordances of shared models in team-based design. In Design thinking (pp. 61-79). Springer, Berlin, Heidelberg.

(ii) There is a huge range of methods to implement creative ideas, and research on the impact of different implementation strategies. Building tangible prototypes is known to be delicate, especially when this requires physical building skills and thereby draws attention away from the ideas at stake. Alternative approaches include storytelling, role-playing, drawing sketches, wizard-of-oz-prototyping, building critical-function-prototypes or designing news-of-the-future. Whenever a skill-intensive form of implementation is required from people, thorough training in the specific implementation method is recommended beforehand, because only with the necessary skill for expressing ideas in a particular medium people can be free to express any kind of an idea freely. The two sample elephants from this study required a notable amount of paperwork skill to be built like this.

Refs:

McKim, R. H. (1972). Experiences in visual thinking. Wadsworth Publishing, Belmont.

d.school (2010) Bootcamp bootleg. https://hpi.de/fileadmin/user_upload/fachgebiete/d-school/documents/01_GDTW-Files/bootcampbootleg2010.pdf

Meinel, C., Rhinow, H., & Köppen, E. (2013). Design Thinking Prototyping Cardset. Hasso Plattner Institut für Softwaresystemtechnik, Potsdam.

Sadler, J., Shluzas, L., Blikstein, P., & Katila, R. (2016). Building blocks of the maker movement: Modularity enhances creative confidence during prototyping. In Design Thinking Research (pp. 141-154). Springer, Cham.

(iii) Even when the time for implementing a creative idea remains the same, there is a huge difference between the implementation as a “one-shot-operation” versus a process of “building and testing, building and testing…”. Via repeated tests and prototype improvements, even implementations that start with little efficacy can be rendered high-performing over time. Therefore, the prospect of being able to test and improve designs throughout the implementation phase allows people to select new and daring ideas, because they can be rendered feasible prior to the final submission. By contrast, when the implementation is (expected to be) a one-time operation, this indeed offers little hope for improvements over time, so that well known and easy to implement concepts seem obviously preferable. In this particular study, no hint has been made to the possibility of tests and refinements in the implementation phase, thereby rendering a “play-safe strategy” a logically good choice for the production of somewhat effective prototypes.

Ref:

Dow, S. & Klemmer, S. R. (2011). The efficacy of prototyping under time constraints. In Design thinking (pp. 111-128). Springer, Berlin, Heidelberg.

(iv) There can be a great difference between the instruction to implement an idea, versus the explicit instruction to implement varying ideas in parallel to probe a bandwidth of creative opportunities. In the latter case, there is also the possibility of instructing students to select at least one idea that seems well-feasible, and another idea that is highly original and would seem to yield very good effects if only it could be implemented (“dark horse” method). In this particular study, it seems to be the case that children were instructed to select (and implement?) two ideas. However, there was obviously no instruction to use this parallel prototyping opportunity in order to enhance more diverse thinking, including the implementation of at least one more extreme and daring idea.

Refs:

Dow, S. P., Glassco, A., Kass, J., Schwarz, M., Schwartz, D. L. & Klemmer, S. R. (2010). Parallel prototyping leads to better design results, more divergence, and increased self-efficacy. Transactions on Computer-Human Interaction, 17 (4), 18:1-24.

Dow, S., Fortuna, J., Schwartz, D., Altringer, B., Schwartz, D., & Klemmer, S. (2011, May). Prototyping dynamics: sharing multiple designs improves exploration, group rapport, and results. In Proceedings of the SIGCHI conference on human factors in computing systems (pp. 2807-2816).

Dow, S. (2011). How prototyping practices affect design results. Interactions, 18(3), 54-59.

https://www.designthinking-methods.com/en/4Prototypen/darkhorse.html

In their paper, the authors need to be very careful in interpreting their results. This experiment has not tested the impact of students expecting to implement or even expecting to construct creative ideas in general terms. It has tested a very specific implementation procedure that aims at (i) refined prototypes, which (ii) require considerable building skills, (iii) constructed in a “one-time” implementation procedure, (iv) without using parallel designs as a means to foster divergent, creative thinking in the implementation phase. Educators should rather be recommended to delve deeper into the existing literature on how different implementation strategies impact creative thinking, rather than being warned about implementation steps as creating a bias against creativity in general terms.

Specific considerations:

In their discussion of practical implications, the authors currently write: “In stimulating children’s creativity, it may be worthwhile for educators to […] guide children with highly original but seemingly unrealistic ideas to think of ways for the idea to be made feasible.” In this context, it can be important to note that this question does not require answers from scratch to be developed in a case-by-case approach, let alone by the students. There is strong empirical data on strategies how to render original ideas more effective in the implementation phase. This could be taught by educators in the first place. A highly effective approach to render original ideas more effective is iterative prototype testing, moving from rough prototypes (concept-based) to more refined prototypes over time.

Furthermore the authors posit: “In stimulating children’s creativity, it may be worthwhile for educators to support children in their intuitive judgments of original ideas […]”. In this regard, again there are strategies that educators can use to help students pursue original ideas. In particular, parallel prototyping with the explicit instruction to choose at least one daring, wild “dark horse” solution is an empirically substantiated method to ensure that highly original ideas do not get lost in the implementation phase.

Second, the authors of the paper have compiled a great battery to assess control variables. However, in this regard the statistical analysis seems not very sensitive. It yields the highly surprising outcome that no control variable at all has a notable impact on the children’s choices of ideas, not even “risk preference” as measured with the Risk Taking 10-item scale from the Jackson Personality Inventory. The authors write: “Notably, the detrimental effect of instructing children to transform their ideas into tangible and physical products was consistent among children with different background characteristics (i.e., gender, age, ethnicity, socioeconomic status, prior level of achievement, school) and psychological characteristics (i.e., risk preference and personality traits). Hence, these findings are broadly applicable to children independent of demographic or psychological variables (S2).” Here, a more pinpointed statistical analysis would seem helpful. For instance, what is the average originality of selected ideas in the 25% of children that are most risk-averse compared to the 25% who are most open to risk-taking? After all, later on the authors wish to conclude:

“This means that by focusing on end-products only, educators may run the risk of applauding the creativity of a small group of children who had the courage to try out original ideas, while at the same time failing to recognize the creative potential of a group of equally creative children who were perhaps too afraid to bring their ideas into practice.” This statement seems to imply that there IS a differential impact of instruction on students. Especially the more cautious students would choose feasible solutions, while the more daring students might still implement highly original ideas.

“Further, it is important for teachers to foster a psychologically safe classroom environment where children feel safe in playing and experimenting with ideas and materials, taking sensible risks, and making mistakes.“ Again, this passage suggests that there IS a differential impact of psychological attributes, such that students who feel safe (e.g., to fail) will be in a better position to develop their creativity, from initial ideation up to the implementation of final project outcomes.

Overall, especially in the context of education, it would be good to show the differential impact of psychological variables such as willingness to take risks on creative performance. After all, education could help to foster a mindset that helps students be creative, e.g. by being better able to cope with the uncertainty of trying out something that is so unique that it has not been tried by anyone before.

Thirdly, it would be good to increase the conceptual clarity regarding the relationship of “originality” and “usefulness”, the two key dimensions characterising creative products. It is not logically necessary that highly original ideas must be less useful / feasible than non-original ideas. In this sense, the Idea Assessment Probes have been constructed as ideas sets for testing purposes in creativity research. These sets include original and non-original ideas as well as useful/feasible and not useful/feasible ideas, with the two dimensions varying independently of each other. Even with these idea sets, empirical research has found that people concerned with idea implementation (in a managerial context) preferred non-original ideas. This was the case even though highly original ideas would have been as feasible. Therefore, the feasibility of implementation does not seem to be the only factor that drives people to shy away from original ideas in the idea selection phase.

Ref:

von Thienen, J. P. A., Ney, S. & Meinel, C. (2019). Estimator socialization in design thinking: The dynamic process of learning how to judge creative work. In R. Beghetto and G. E. Corazza (eds.), Dynamic perspectives on creativity: New directions for theory, research and practice in education (pp. 67-99). Springer.

In the present paper, it would be interesting to read the statistical correlation of idea "originality" and "usefulness" ratings by the children and the experts.

Moreover, it would be interesting to read if there might be a preference for unoriginal ideas that is even independent of feasibility considerations (e.g., a negative correlation of “originality” with “choice for implementation”, while partialising out the “feasibility” rating).

Forthly, the paper discusses the impact of “conformist thinking” in children. For instance: “several studies show that children around this age begin a trend of increasingly conformist thinking that continues through high school (20-22). Accordingly, it can be expected that our findings may become even stronger among young adolescents.“ Here it is not clear why conformist thinking should lead to a bias against original ideas. After all, the role models with whom children/adolescents might want to conform could be teachers or peers who explicitly love wild and daring creative ideas. Maybe the logic of argumentation could be explained in more detail here. Alternatively, there might be a discussion regarding the importance of role models and the impact of educators as role models in this context.

All in all, this paper conveys a lot of important information in a condensed format; it is written in good English and very understandable. The authors master a good integration of a controlled experimental approach in an ecologically valid school setting, with a large sample size and a sophisticated testing battery. It is a very good piece of research, definitely recommended for publication. The required revisions range between minor and major. In particular, the theoretical background, interpretation and conclusion warrant further attention. Most importantly, varying methods exist for the implementation of creative ideas. Research has already demonstrated that these methods/strategies have hugely varying impacts on creative outcomes.

6. PLOS authors have the option to publish the peer review history of their article (what does this mean?). If published, this will include your full peer review and any attached files.

Reviewer #1: No

Reviewer #2: No

---

## [Decision Letter · Decision Letter 1]

5 Jul 2022

Instructing children to construct ideas into products alters children's creative idea selection in a randomized field experiment

PONE-D-22-04429R1

Dear Dr. van Broekhoven,

We’re pleased to inform you that your manuscript has been judged scientifically suitable for publication and will be formally accepted for publication once it meets all outstanding technical requirements.

Kind regards,

Sergio Agnoli

Academic Editor

PLOS ONE

Additional Editor Comments (optional):

Congratulations on your nice work and on the careful revision. I agree with both reviewers that the work is now ready for publication in Plos One. As a last suggestion, I would recommend to take into account the suggestion by Reviewer 1 for the rephrasing of the highlighted sentence (during the proofs correction stage). Good luck for you future work!

Reviewers' comments:

Reviewer's Responses to Questions

**Comments to the Author**

1. If the authors have adequately addressed your comments raised in a previous round of review and you feel that this manuscript is now acceptable for publication, you may indicate that here to bypass the “Comments to the Author” section, enter your conflict of interest statement in the “Confidential to Editor” section, and submit your "Accept" recommendation.

Reviewer #1: All comments have been addressed

Reviewer #2: All comments have been addressed

2. Is the manuscript technically sound, and do the data support the conclusions?

Reviewer #1: Yes

Reviewer #2: Yes

3. Has the statistical analysis been performed appropriately and rigorously? 

Reviewer #1: Yes

Reviewer #2: Yes

4. Have the authors made all data underlying the findings in their manuscript fully available?

Reviewer #1: Yes

Reviewer #2: No

5. Is the manuscript presented in an intelligible fashion and written in standard English?

Reviewer #1: Yes

Reviewer #2: Yes

6. Review Comments to the Author

Reviewer #1: The authors have done a great job in revising their work. All of the initially raised issues were clearly addressed and I thank the authors for their responsiveness. As this is an even better version of an already nice paper, I am happy to recommend it for publication in PLOS ONE.

Just one very small suggestion: Seems like the following sentence needs a slight rephrasing: "Hence, both untreated and treated conscientiousness children show the behavior of others when they are treated."

Reviewer #2: Thank you for the careful consideration of feedback, the line of argumentation now seems strong and consistent. Looking forward to seeing this article in print.

7. PLOS authors have the option to publish the peer review history of their article (what does this mean?). If published, this will include your full peer review and any attached files.

Reviewer #1: No

Reviewer #2: No

---

## [Editor Report · Acceptance letter]

26 Jul 2022

PONE-D-22-04429R1 

Instructing children to construct ideas into products alters children’s creative idea selection in a randomized field experiment 

Dear Dr. van Broekhoven:

I'm pleased to inform you that your manuscript has been deemed suitable for publication in PLOS ONE. Congratulations! Your manuscript is now with our production department. 

Kind regards, 

on behalf of

Dr. Sergio Agnoli 

Academic Editor

PLOS ONE